# A neuronal mechanism underlying decision-making deficits during hyperdopaminergic states

Jeroen P.H. Verharen [1,2], Johannes W. de Jong[1,3], Theresia J.M. Roelofs[1], Christiaan F.M. Huffels[1], Ruud van Zessen[1], Mieneke C.M. Luijendijk[1], Ralph Hamelink[4,5], Ingo Willuhn[4,5], Hanneke E.M. den Ouden [6], Geoffrey van der Plasse[1], Roger A.H. Adan[1] & Louk J.M.J. Vanderschuren[2]

Hyperdopaminergic states in mental disorders are associated with disruptive deficits in decision making. However, the precise contribution of topographically distinct mesencephalic dopamine pathways to decision-making processes remains elusive. Here we show, using a multidisciplinary approach, how hyperactivity of ascending projections from the ventral tegmental area (VTA) contributes to impaired flexible decision making in rats. Activation of the VTA–nucleus accumbens pathway leads to insensitivity to loss and punishment due to impaired processing of negative reward prediction errors. In contrast, activation of the VTA–prefrontal cortex pathway promotes risky decision making without affecting the ability to choose the economically most beneficial option. Together, these findings show how malfunction of ascending VTA projections affects value-based decision making, suggesting a potential mechanism through which increased forebrain dopamine signaling leads to aberrant behavior, as is seen in substance abuse, mania, and after dopamine replacement therapy in Parkinson's disease.

[1] Department of Translational Neuroscience, Brain Center Rudolf Magnus, University Medical Center Utrecht, Universiteitsweg 100, 3584 CG Utrecht, The Netherlands. [2] Department of Animals in Science and Society, Division of Behavioural Neuroscience, Faculty of Veterinary Medicine, Utrecht University, Yalelaan 2, 3584 CM Utrecht, The Netherlands. [3] Department of Molecular and Cell Biology and Helen Wills Neuroscience Institute, University of California, Berkeley, 132 Barker Hall, Berkeley, CA 94720, USA. [4] Netherlands Institute for Neuroscience, Institute of the Royal Netherlands Academy of Arts and Sciences, Meibergdreef 47, 1105 BA Amsterdam, The Netherlands. [5] Department of Psychiatry, Academic Medical Center, University of Amsterdam, Meibergdreef 9, 1105 AZ Amsterdam, The Netherlands. [6] Donders Institute for Brain, Cognition and Behavior, Radboud University, Montessorilaan 3, 6525 HR Nijmegen, The Netherlands. Roger A. H. Adan and Louk J. M. J. Vanderschuren contributed equally to this work. Correspondence and requests for materials should be addressed to R.A.H.A. (email: r.a.h.adan@umcutrecht.nl) or to L.J.M.J.V. (email: l.j.m.j.vanderschuren@uu.nl)

Impaired decision making can have profound negative consequences, both in the short and in the long term. As such, it is observed in a variety of mental disorders, such as mania[1,2], substance addiction[3–6], and as a side effect of dopamine (DA) replacement therapy in Parkinson's disease[7,8]. Importantly, these disorders are associated with aberrations in DAergic neurotransmission[9,10], and DA has been implicated in decision-making processes[11–13]. However, ascending DAergic projections from the ventral mesencephalon are anatomically and functionally heterogeneous[14–16] and the contribution of these distinct DA pathways to decision-making processes remains elusive.

The mesocorticolimbic system, comprising DA cells within the ventral tegmental area (VTA) that mainly project to the nucleus accumbens (NAc; mesoaccumbens pathway) and medial prefrontal cortex (mPFC; mesocortical pathway), has an important role in value-based learning and decision making[14–16]. When an experienced reward is better than expected, the firing of VTA DA neurons increases, thereby signaling a discrepancy between anticipated and experienced reward to downstream regions. Conversely, when a reward does not fulfill expectations, DA neuronal activity decreases. This pattern of DA cell activity is the basis of reward prediction error (RPE) theory[17–20], which describes an essential mechanism through which organisms learn to flexibly alter their behavior when the costs and benefits associated with different courses of action shift. Although the relevance of RPEs in value-based learning is widely acknowledged, little is known about how different VTA target regions process these DA-mediated error signals, and how this ultimately leads to adaptations in behavior.

Here, we used projection-specific chemogenetics combined with behavioral tasks, pharmacological interventions, computational modeling, in vivo microdialysis, and in vivo neuronal population recordings to investigate how different ascending VTA projections contribute to value-based decision-making processes in the rat. Specifically, we investigated the mechanism underlying the aberrant decision-making style that is associated with increased DA neuron activity. We hypothesized that hyperactivation of VTA neurons interferes with reward prediction error processing, leading to impaired adaptation to reward value dynamics. We predicted an important contribution of the mesoaccumbens pathway in incorporating experienced reward, loss, and punishment into future decisions, considering the importance of the NAc in reinforcement learning and motivated behaviors[21–23], and a modulatory role for the mesocortical pathway in value-based choice behavior, given its involvement in executive functions, such as decision making and behavioral flexibility[24,25]. Furthermore, we tested an explicit prediction based on a neurocomputational model of the DA system, in which impaired negative RPE processing is involved in learning deficits during DA replacement therapy[7,26]. Our data show that activation of the VTA–NAc pathway reduces the sensitivity to loss and punishment as a result of impaired processing of negative RPEs, whereas activation of the VTA–mPFC pathway promotes risky decision making, but only when this entails no loss of reward. Together, these findings shed light on the behavioral mechanisms by which increased activity of distinct ascending DA projections contributes to deficits in value-based decision-making processes.

## Results

**Dopaminomimetic drugs impair serial reversal learning.** To test the role of DA in flexible value-based decision making, rats were tested in a serial reversal learning task following systemic treatment with the DA neurotransmission enhancers cocaine and D-amphetamine. A reversal learning session (Fig. 1a) comprised 150 trials, and started with the illumination of two nose poke holes in an operant conditioning chamber. One of these was randomly assigned as active, and responding in this hole resulted in sucrose delivery under a fixed ratio (FR) 1 schedule of reinforcement. When animals had made five consecutive correct responses, the contingencies reversed so that the previously inactive hole now became active, and vice versa.

Injection of either drug did not affect the number of trials needed to reach the criterion of a series of five consecutive correct responses (Fig. 1b, left panel). However, the number of reversals achieved in the entire session was significantly reduced in the drug-treated animals (Fig. 1b, right panel, and Supplementary Fig. 1a). Thus, cocaine and D-amphetamine impaired task performance, but this effect did not appear until the moment of first reversal. We reasoned that this pre- and post-reversal segregation in drug effects on task performance is related to the structure of the task (Fig. 1a). That is, after every reversal, the value of the outcome of responding in the previously active hole declines and, conversely, the value associated with responding in the previously inactive hole increases. Accordingly, this task entails a combination of devaluation and revaluation mechanisms following reversals.

To understand the nature of the drug-induced deficit in reversal learning performance, we analyzed the animals' behavior in more detail. Perseverative responding, i.e., the average number of responses in the previously active hole directly after a reversal, was not altered after cocaine or D-amphetamine treatment (Fig. 1c). Lose-stay behavior, i.e. the percentage of (unrewarded) trials in the inactive nose poke hole followed by a response in the (still) inactive hole, was also not affected (Fig. 1d, left panel). However, win-stay behavior, i.e., the percentage of responses in the active nose poke hole after which the animal responded in that same active hole, was significantly decreased after treatment with cocaine or D-amphetamine (Fig. 1d, right panel). This drug-induced reduction in win-stay behavior indicates that even though the animals received a reward after responding in the active nose poke hole, they next sampled the inactive hole more often than after saline treatment. Importantly, win-stay behavior was only reduced after reversal, indicating that behavioral impairments were not the result of a general decline in task performance or sensitivity to reward.

Overall, the effects in the reversal learning task indicate that increased DA signaling after cocaine or D-amphetamine treatment did not impair the animals' ability to find the active nose poke hole at task initiation, hence to assign positive value to an action. Yet, when the values of (the outcome of) two similar actions (that is, responding in a nose poke hole) changed relative to each other, drug-treated animals were impaired in adjusting behavior, perhaps as a result of a valuation deficit. This suggests that treatment with these drugs disrupted the process of integrating recent wins or losses (i.e., a revaluation or a devaluation impairment, respectively) in decisions.

To gain insight into the mechanisms underlying impaired reversal learning, we modeled the behavior of each subject by fitting the data to a computational reinforcement learning model (Fig. 1e, f and Supplementary Table 1). We used an extended version of the Rescorla–Wagner model[27,28], using two different learning rates, $a_{win}$ and $a_{loss}$, describing the animal's ability to learn from wins and losses, respectively[29]. Such a model-based approach investigates task performance based on an extended history of trial outcomes, and not merely the most recent outcome, such as win- and lose-stay measures do, providing a more in-depth analysis of the learning capacity of the animals.

When comparing the Rescorla–Wagner model coefficients of the animals after saline with those after cocaine and D-amphetamine treatment, we observed a strong decrease in

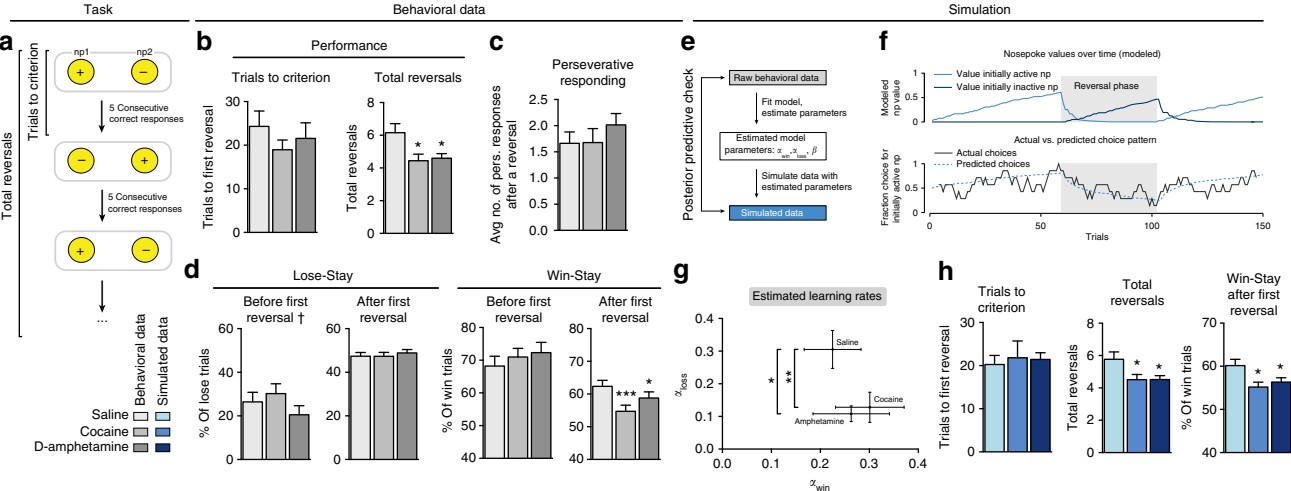

**Fig. 1** Treatment with cocaine or D-amphetamine impairs reversal learning. **a** Task design. **b** Systemic cocaine (10 mg/kg) or D-amphetamine (0.25 mg/kg) treatment did not alter the number of trials required to reach first reversal (one-way RM ANOVA, $p = 0.55$), but decreased the total number of reversals accomplished (ANOVA, $p = 0.0037$; post-hoc test versus saline, $p = 0.0102$ cocaine, $p = 0.0197$ D-amphetamine). **c** Treatment with cocaine or D-amphetamine did not alter perseverative behavior after a reversal ($p = 0.46$). **d** Lose-stay behavior was unaffected after cocaine or D-amphetamine treatment, both before ($p = 0.21^{†}$) and after ($p = 0.77$) the first reversal. Cocaine and D-amphetamine decreased win-stay behavior after (ANOVA, $p = 0.0007$; post-hoc test versus saline, $p = 0.0009$ for cocaine, $p = 0.0336$, D-amphetamine), but not before the first reversal ($p = 0.67$). Repeated measures from $n = 25$ animals. $^{†}$Six animals had no losses before the first reversal, so the ANOVA was performed on data of $n = 19$ animals; graph shows n = 25. **e** We used a modified Rescorla–Wagner model to describe the behavior of the rats during reversal learning. **f** Example session. (Top) Simulated values of the nose pokes, given the rat's optimal model parameters and observed choices. (Bottom) Modeled choice probabilities, converted from the simulated nose poke values using a softmax (unsmoothed), and the rat's actual choice pattern (smoothed over 7 trials). **g** Best-fit learning parameters. Treatment with cocaine and D-amphetamine decreased $\alpha_{loss}$, without affecting the other model coefficients (Wilcoxon matched-pairs signed rank test, $^*p = 0.032$, $^{**}p = 0.0046$, see Supplementary Table 2). **h** Simulating data with the model parameters extracted in **g** replicated the drug-induced effects of the behavioral data ($n = 25$ simulated rats; ANOVA trials to criterion, $p = 0.86$; total reversals, $p = 0.0114$, post-hoc test, $p = 0.0411$ for cocaine, and $p = 0.0215$ for D-amphetamine; ANOVA win-stay, $p = 0.0090$, post-hoc test, $p = 0.0181$ for cocaine and $p = 0.0462$ for D-amphetamine. ANOVA on all other measures, $p > 0.1$). Data shown as mean ± standard error of the mean; $^*p < 0.05$, $^{**}p < 0.01$, $^{***}p < 0.001$

parameter $a_{loss}$ without affecting $a_{win}$ or choice stochasticity factor β (Fig. 1g, h, Supplementary Fig. 1b, c and Supplementary Table 2). This indicates that cocaine and D-amphetamine interfere with learning from negative, but not positive, RPEs.

**Mesoaccumbens pathway activation impairs reversal learning.** In view of the role of DA in RPE signaling, we hypothesized that cocaine and D-amphetamine interfered with learning from losses by overactivation of ascending midbrain DA projections, thereby disrupting negative RPEs. This same mechanism has been hypothesized to be involved in the DA dysregulation syndrome in medicated Parkinson's disease patients[7,30]. Such an overactivation may lead to an inability to devalue stimuli and/or their associated outcomes, resulting in choice behavior that is not optimally value based. Specifically, we were interested in the contribution of projections from the VTA to the NAc and the mPFC to impairments in reversal learning.

In order to activate neuronal subpopulations of the VTA in a projection-specific manner, we combined a canine adeno-associated virus retrogradely delivering Cre-recombinase (CAV2-Cre) and a Cre-dependent viral vector encoding hM3Dq(Gq)-DREADD fused to mCherry-fluorescent protein[31] (Fig. 2a and Supplementary Fig. 2). This two-viral approach resulted in high levels of DA specificity (80% of the transfected neurons in the mesoaccumbens group and 72% of the transfected neurons in the mesocortical group were positive for tyrosine hydroxylase, Fig. 2b). To investigate whether the effects of cocaine and D-amphetamine on reversal learning were driven by activation of the mesoaccumbens or mesocortical pathway,

animals were injected with clozapine-N-oxide (CNO) immediately before testing in the reversal learning task.

Chemogenetic activation of the mesoaccumbens pathway resulted in the same pattern of impairments in reversal learning as cocaine and D-amphetamine treatment, i.e., a reduction in the numbers of reversals achieved, without affecting trials to first reversal criterion (Fig. 2c). This pattern was confirmed by plotting the cumulative reversals as a function of completed trials (Fig. 2d and Supplementary Fig. 3a). Similar to cocaine and D-amphetamine, the performance impairment during mesoaccumbens activation was associated with a post-reversal (but not pre-reversal) decrease in win-stay behavior (Fig. 2e), whereas perseverative responding and lose-stay behavior were not altered (Fig. 2f and Supplementary Fig. 3b). Remarkably, during mesoaccumbens activation, both win- and lose-stay behavior were around 50% post reversal, indicative of random choice behavior. Indeed, the Rescorla–Wagner model fitted with a significantly lower likelihood after mesoaccumbens activation (Supplementary Fig. 3c), indicating that the animals' performance declined such that the model was less able to describe the data compared to baseline conditions. In contrast to mesoaccumbens activation, mesocortical activation or CNO injection in a sham-operated control group had no effect on reversal learning.

The finding that hyperactivity in the mesoaccumbens pathway evoked similar effects on reversal learning as cocaine and D-amphetamine did suggest that these drugs exert their influence on flexible value-based decision making through DA neurotransmission within the NAc. To directly test this, we performed in vivo microdialysis in the NAc of animals that expressed Gq-DREADD in the mesoaccumbens pathway (Fig. 2g). Administration of CNO

increased baseline levels of DA in the NAc, as well as its metabolites 3,4-dihydroxyphenylacetic acid (DOPAC) and homovanillic acid (HVA) (Fig. 2h and Supplementary Fig. 4). Next, we infused the DA receptor antagonist α-flupenthixol into the NAc of DREADD-treated animals prior to chemogenetic activation of

the mesoaccumbens pathway in a reversal learning test (Fig. 2i). This dose of α-flupenthixol had no effect after systemic saline injection, but it prevented the effect of chemogenetic activation of the mesoaccumbens pathway on reversal learning (Fig. 2j). This finding supports the assumption that the effects of

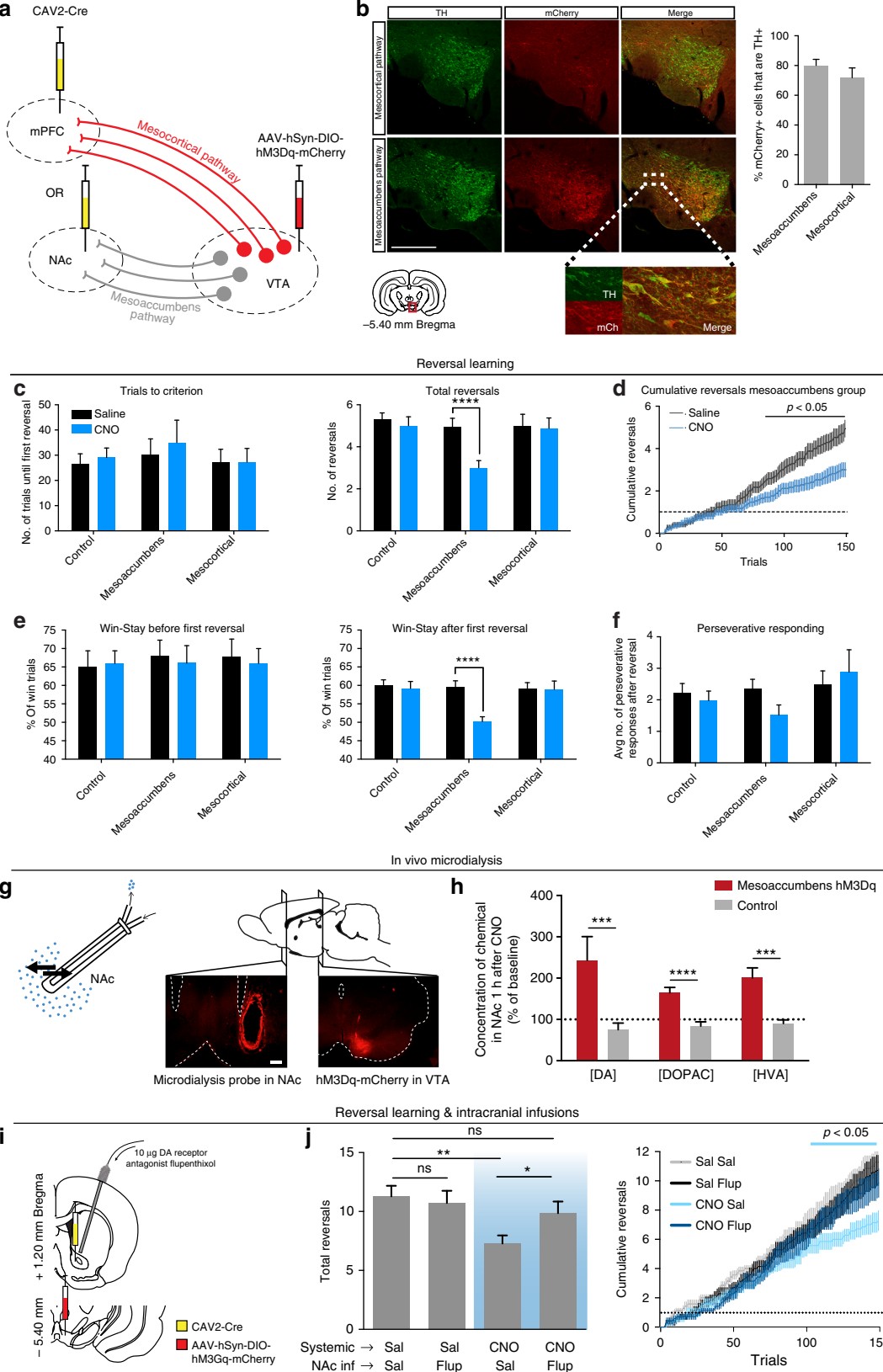

mesoaccumbens hyperactivity are mediated through NAc DA receptor stimulation.

**Dopamine neuron activity during reversal learning.** Considering the function of RPEs in value updating[20], we tested whether midbrain DA neurons tracked the presence of wins and losses in the form of RPEs during reversal learning. To this aim, we measured in vivo neuronal population activity from DA neurons in the VTA using fiber photometry[32] in TH::Cre rats (Fig. 3a and Supplementary Movie 1).

Around the time of responding, we observed a clear two-component RPE signal[20] (Fig. 3b, c and Supplementary Fig. 5), i.e., a ramping of DA activity towards the moment of response, followed by an additional value component. That is, win trials were associated with a prolonged DA peak, whereas loss trials were characterized by a rapid decline in DA population activity after the response was made. No such signals were observed in animals injected with an activity-independent control fluorophore (Supplementary Fig. 5).

Since mesoaccumbens hyperactivity only affected task performance after reversal, we compared DA activity before and after reversal (Fig. 3c, right panels). In loss trials, we observed significantly stronger negative RPEs after the first reversal compared to before reversal. In contrast, DA peaks during the win trials were similar before and after the first reversal. This supports our notion that the impairment in reversal learning during mesoaccumbens hyperactivity was due to selective interference with learning from negative RPE-guided feedback.

**Mesoaccumbal activation impairs adapting to devaluation.** To examine whether the effects of mesoaccumbens hyperactivity on learning from negative feedback generalizes to conditions beyond reversal learning, we trained rats on a probabilistic discounting task (modified from refs. [33,34]). In this task, rats could choose between responding on a 'safe' lever, which always produces one sucrose pellet, or on another, 'risky' lever, which produces a larger reward (i.e., three sucrose pellets) with a given probability. Within a session, the chance of receiving the large reward after a response on the risky lever decreases across four trial blocks—in the first block, animals always received the large reward when pressing the risky lever, whereas the odds of winning were reduced to 1 in 12 in the fourth block (Fig. 4a and Supplementary Fig. 6a). An important difference with reversal learning is that in this task, a response shift is not the best option after a loss per se—lose-stay behavior at the risky lever may yield the same amount of sucrose as a shift to the safe lever, depending on the odds in the trial block. Therefore, an increase in lose-stay or decrease in win-stay behavior does not necessarily reflect poor choice behavior.

After training, the animals showed stable discounting performance, preferring the risky lever in the first block, and shifting their choice towards the safe lever when the yield of the risky lever diminished (Fig. 4b, left panel). Mesoaccumbens activation (Fig. 4b, middle panel) decreased the choice of the risky lever in the first block and increased choice for the risky lever in the last block, resulting in a significantly reduced slope of the discounting curve (Fig. 4b, middle panel, inset), and a lower percentage of optimal choices (Fig. 4c). Importantly, the inability to discount the value of the risky lever in the latter blocks of the task is indicative of an inability to adapt to a declining outcome of responding on the risky lever (Supplementary Fig. 6b). The reduced choice for the risky lever in the first block may also be due to a devaluation deficit, as the receipt of only one sucrose pellet after responding on the safe lever (compared to the three-pellet yield of responding on the risky lever) may be perceived as a 'loss', since the relative value of responding on the safe lever is lower in this block[35]. In contrast, mesocortical activation only increased risk seeking in the second block, in which the yield of responding on the safe (1 pellet) and risky (1 in 3 chance of 3 pellets) levers was equal (Fig. 4b, right panel), so that the amount of optimal choices remained unaffected (Fig. 4c). Further analysis of task strategy showed that lose-stay behavior at the risky lever was increased during activation of the mesoaccumbens and mesocortical pathways, whereas win-stay and safe-stay behavior were unaffected (Fig. 4d and Supplementary Fig. 6c). Thus, activation of both ascending VTA projections made animals less prone to alter choice behavior after losses, which significantly impaired task performance during mesoaccumbens activation. The increase in lose-stay behavior during mesocortical activation is the result of the preference for the risky lever in the second trial block, but this did not result in poor choice behavior (Fig. 4c).

To test whether the effects in this task were specific to devaluation mechanisms, we trained the animals expressing DREADD in mesoaccumbens neurons on the same task with increasing instead of decreasing odds of reward at the risky lever (Fig. 4e). In this condition, mesoaccumbens activation did not significantly change risky choice in any of the blocks (Fig. 4f), although a modest but significant decrease was observed in performance (i.e., a lower fraction of optimal choices; Fig. 4g) which was caused by a higher preference for the risky lever in the first few trials (Supplementary Fig. 6d). This could be the result of a reduced ability of the animals to devalue the outcome of responding on the risky lever in the initial trials of the first block. However, since this version of the task primarily relies on revaluation, rather than devaluation mechanisms, especially in later blocks (Supplementary Fig. 6b), a mesoaccumbens stimulation-induced devaluation deficit caused no further changes in behavior. Indeed, win-stay and lose-stay behavior were unaffected by mesoaccumbens activation (Fig. 4g).

In sum, the effects of chemogenetic activation in the probabilistic discounting task support our hypothesis that

**Fig. 2** Chemogenetic activation of the mesoaccumbens, but not mesocortical, pathway mimicked the effects of cocaine and D-amphetamine on reversal learning. **a** Experimental design. **b** (Left panel) Representative histology images showing coronal sections stained for tyrosine hydroxylase (left), DREADD-mCherry (middle), and an overlay (right). Scale bar, 500 μm. (Right panel) Co-staining of mCherry with tyrosine hydroxylase, showing the percentage of DREADD-transfected neurons that is dopaminergic (mean ± s.d.). Data from $n = 9$ (mesoaccumbens), $n = 8$ (mesocortical) animals. **c–f** Chemogenetic mesoaccumbens stimulation mimicked the effects of cocaine and D-amphetamine on reversal learning. All data: $n = 17$ control, $n = 17$ mesoaccumbens, $n = 16$ mesocortical group; ****$p < 0.0001$ in post-hoc test. See Supplementary Table 3. **g** Microdialysis was used to measure extracellular concentrations of DA and its metabolites in the NAc after chemogenetic mesoaccumbens stimulation. Scale bar, 500 μm. **h** NAc levels of DA and its metabolites were elevated 1 h after an i.p. CNO injection in DREADD-infected animals compared to controls (post-hoc tests, DA, $p = 0.0002$; DOPAC, $p < 0.0001$; HVA, $p = 0.0008$; ***$p < 0.001$, ****$p < 0.0001$; see also Supplementary Fig. 4). **i** Prior to reversal learning, animals received systemic CNO (or saline) for DREADD stimulation and a microinjection with α-flupenthixol (or saline) into the nucleus accumbens. **j** α-Flupenthixol itself had no effect on reversal learning, but prevented the CNO-induced impairment on reversal learning (ANOVA, $p = 0.0024$; post-hoc test: **$p = 0.0019$, *$p = 0.0397$). Note that animals had a higher baseline of reversals in this experiment, because the animals were trained on the task (see Methods). Sal saline, Flup α-flupenthixol, ns not significant

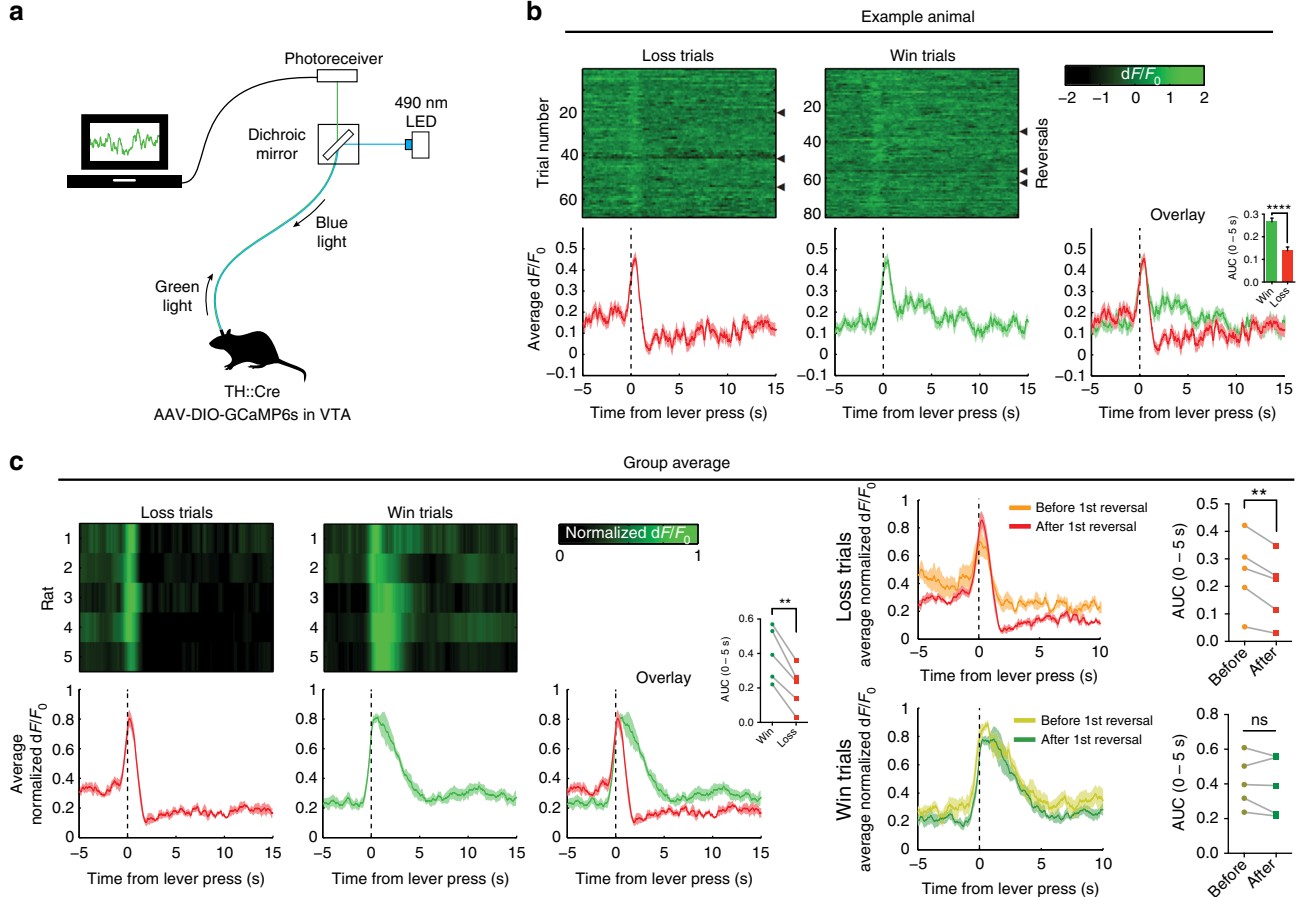

**Fig. 3** In vivo fiber photometry in VTA DA neurons during reversal learning. **a** Experimental setup. **b** Reversal learning session of an example animal. Triangles depict a reversal. Data are time locked to a lever press by the rat and (in win trials) immediate reward delivery. Inset shows area under the curve in the first 5 s following lever press (unpaired $t$-test, $p < 0.0001$). **c** Group average. (Left panels) VTA DA neurons responded differentially to wins and losses (AUC (inset), paired t-test, $p = 0.0015$). (Right panels) Loss trials evoked a stronger negative reward prediction error signal after the first reversal compared to before reversal. (AUC (inset), paired $t$-test, $p = 0.0062$ for loss trials, $p = 0.3658$ for win trials); ns not significant, **$p < 0.01$, ****$p < 0.0001$

mesoaccumbens activation results in an inability of animals to adapt behavior to lower-than-expected outcomes, which under physiological circumstances is mediated by negative RPE signals in DA cells. In contrast, mesoaccumbens hyperactivity did not markedly interfere with adaptations to higher-than-expected outcomes. Furthermore, mesocortical activation increased risky choice behavior, but only when this was without negative consequences for the net gain in the task.

**Dopamine activation does not change static reward value.** Changes in static reward value may influence behavior in tasks investigating dynamic changes in reward value, such as the reversal learning task. For example, food rewards may be less or more appreciated due to changes in feelings of hunger, satiety, or pleasure. Alternatively, operant responding may become habitual rather than goal directed when manipulating the striatum, although this is thought to be mediated by its dorsal parts rather than the NAc[22,36].

To assess whether alterations in static reward value or in the associative structure of operant responding contributed to the behavioral changes evoked by DA pathway stimulation, rats were subjected to operant sessions in which they could lever press for sucrose under an FR10 schedule of reinforcement. Activation of the mesoaccumbens and mesocortical pathways did not alter the total number of lever presses (Fig. 5a), suggesting that absolute

reward value was unchanged. We also tested animals in operant sessions, whereby in half of the sessions the animals were pre-fed with the to-be obtained reward. This type of devaluation tests whether animals retain the capacity to adjust operant behavior to changes in (the representation of) reward value. Prefeeding robustly diminished lever pressing for sucrose, both in a non-reinforced extinction session and under an FR5 schedule of reinforcement. Importantly, this effect of reward devaluation was not affected by mesoaccumbens or mesocortical activation (Fig. 5b), indicating that responding remained goal directed[36].

Consistent with previous findings[37,38], activation of the mesoaccumbens pathway increased operant responding under a progressive ratio schedule of reinforcement[39] (Fig. 5c), which is usually thought to reflect an increased motivation to obtain food[37–39]. However, in light of the present findings, we interpret this finding as that mesoaccumbens hyperactivity renders animals less able to devalue the relative outcome of pressing the active lever when the response requirement increases over the session, hence leading to increased response levels. Such an action devaluation likely involves negative RPE signals from DA neurons.

**Mesoaccumbens hyperactivity evokes punishment insensitivity.** To test whether the devaluation deficit as a result of mesoac-cumbens hyperactivity also resulted in an inability to incorporate

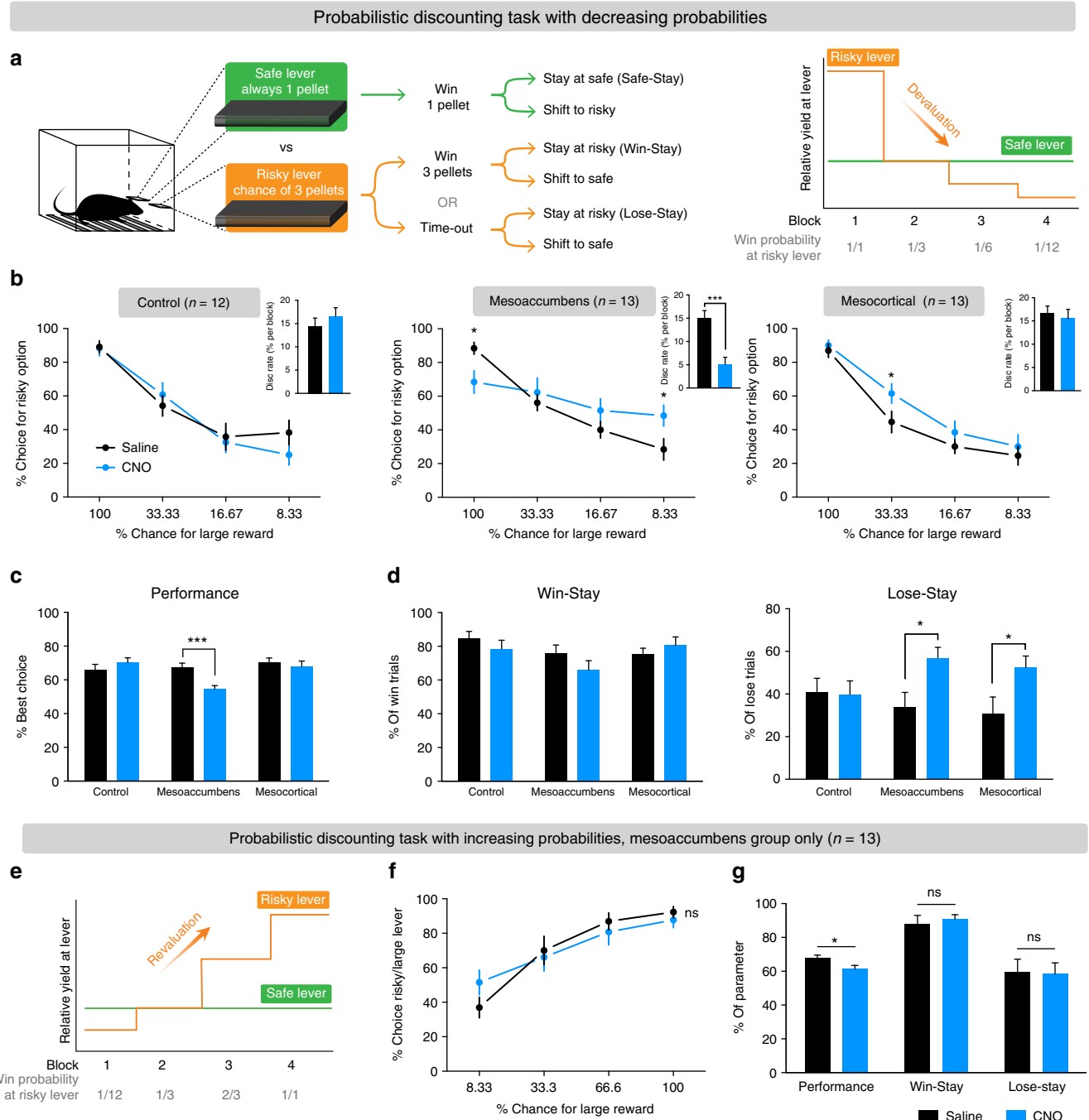

**Fig. 4** Chemogenetic activation of the mesoaccumbens and the mesocortical pathway alters probabilistic discounting. **a** Task design. **b** Discounting curves for individual groups. (Left panel) Sham control group (saline vs CNO; Sidak's test, $p > 0.1$ for all blocks). (Middle panel) During mesoaccumbens hyperactivity, animals have a smaller preference for the risky lever in the first block (Sidak's test, $p = 0.0468$), a larger preference for the risky lever in the last block ($p = 0.0468$; blocks 2 and 3 both $p > 0.1$), and a significantly diminished discounting rate (inset, $p = 0.0002$). (Right panel) Mesocortical activation increased choice for the risky lever in the second block (Sidak's test in block 2, $p = 0.0247$; blocks 1, 3, and 4, all $p > 0.1$). Asterisks in discounting curves indicate significant difference between saline and CNO treatment. Insets display the average steepness of the discounting curve (statistical comparison with Sidak's test). **c** Mesoaccumbens activation reduces the percentage optimal choices in the probabilistic discounting task (i.e., % best choice in blocks 1, 3, and 4; two-way ANOVA; effect of CNO, $p = 0.0331$; group×CNO interaction, $p = 0.0016$; post-hoc Sidak's test, $p = 0.5082$ for control group, $p = 0.0004$ for mesoaccumbens group, $p = 0.7533$ for mesocortical group). **d** Chemogenetic activation of the mesoaccumbens or mesocortical pathway had no effect on win-stay behavior (two-way ANOVA; effect of CNO, $p = 0.36$; group×CNO effect, $p = 0.26$), but did increase lose-stay behavior (two-way repeated ANOVA; effect of CNO, $p = 0.0026$; group×CNO effect, $p = 0.0622$; post-hoc Sidak's test, $p = 0.9998$, $p = 0.0177$ and $p = 0.0203$ for control, mesoaccumbens, and mesocortical groups, respectively). **e** Task design of the probabilistic discounting task with increasing probabilities. **f** Mesoaccumbens activation did not affect the discounting curve (Sidak's test in every block, $p > 0.1$). **g** Mesoaccumbens activation decreased performance on the task (paired $t$-test, $p = 0.0143$), but not win-stay (paired $t$-test, $p = 0.32$) or lose-stay behavior (paired $t$-test, $p = 0.85$). Data are shown as mean ± standard error of the mean; ns not significant, $*p < 0.05$, $***p < 0.001$

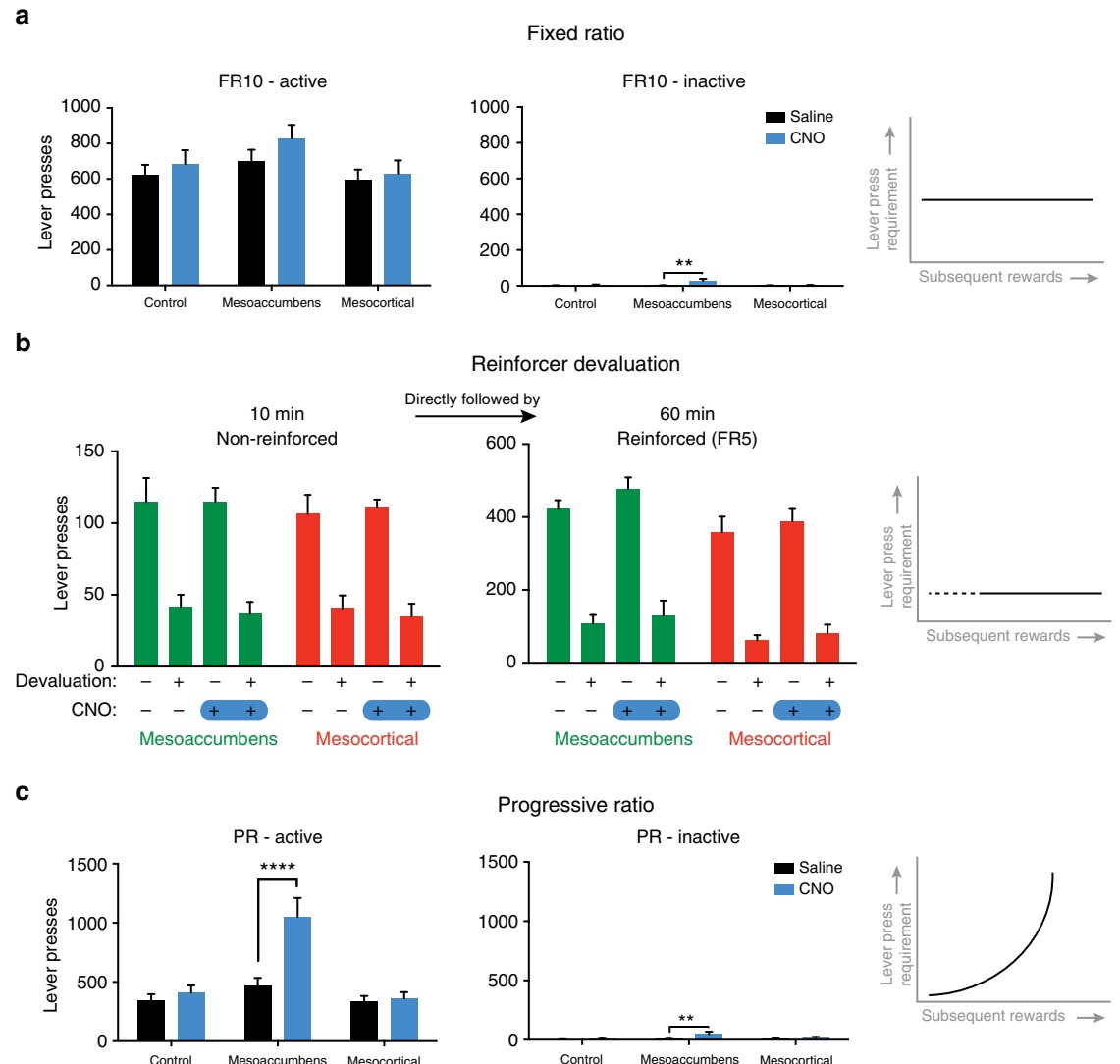

**Fig. 5** Mesocortical and mesoaccumbens activation does not alter the static reward value of sucrose. **a** DREADD activation of either pathway did not affect the number of active lever presses under an FR10 schedule of reinforcement (two-way repeated ANOVA; effect of CNO, $p = 0.0355$; group×CNO effect, $p = 0.5001$; post-hoc test, CNO versus saline, all $p > 0.1$). A modest increase was observed in inactive lever presses after mesoaccumbens activation (two-way ANOVA; effect of CNO, $p = 0.0096$; group×CNO effect, $p = 0.0207$; post-hoc test, CNO versus saline, $p = 0.9302$ controls, $p = 0.0017$ mesoaccumbens group; $p = 0.9957$ mesocortical group); $n = 9$ for control, $n = 8$ for mesoaccumbens, $n = 9$ for mesocortical. **b** Both during a 10 min extinction session (left) and a reinforced lever pressing session (FR5, right), devaluation of the reinforcer by selective satiation for sucrose lead to a decrease in responding (2-way ANOVA, effect of prefeeding in all groups, $p < 0.0001$), without any CNO effects (non-reinforced mesoaccumbens, CNO effect $p = 0.7745$, prefeeding×CNO effect: $p = 0.8448$; non-reinforced, mesocortical, CNO effect $p = 0.9516$, prefeeding×CNO effect: $p = 0.5318$; reinforced mesoaccumbens, CNO effect $p = 0.1472$, prefeeding×CNO effect: $p = 0.5287$; reinforced mesocortical, CNO effect $p = 0.4654$, prefeeding×CNO effect: $p = 0.8877$); $n = 12$ mesoaccumbens, $n = 11$ mesocortical group. **c** Under a progressive ratio schedule of reinforcement, mesoaccumbens activation increased the number of lever presses made (two-way ANOVA; effect of CNO, $p = 0.0006$; group×CNO effect, $p = 0.0007$; post-hoc test, $p = 0.8998$ controls; $p < 0.0001$ mesoaccumbens group; $p = 0.9947$ mesocortical group). A modest increase in inactive lever presses was observed after mesoaccumbens stimulation (two-way ANOVA; effect of CNO, $p = 0.0204$; group×CNO effect, $p = 0.0680$; post-hoc test, CNO versus saline, $p = 0.9840$ controls; $p = 0.0082$ mesoaccumbens group; $p = 0.9392$ mesocortical group); $n = 9$ controls, $n = 8$ mesoaccumbens group, $n = 9$ mesocortical group. Data shown as mean ± standard error of the mean; **$p < 0.01$, ****$p < 0.0001$

explicitly negative consequences into a decision, we subjected animals to a novel punishment task, in which reward taking was paired with an increasing chance of an inescapable foot shock (Fig. 6a). As expected, the introduction of this 0.3 mA foot shock punishment diminished responding for sucrose, an effect that persisted after injection of CNO in the mesocortical and sham control groups (Fig. 6b). In contrast, activation of the mesoaccumbens pathway completely abolished this punishment-induced reduction in responding, as the animals took as many rewards as under non-punishment conditions. This finding suggests that

during mesoaccumbens hyperactivity, reward value is not properly discounted—in other words, animals are not able to take the increasingly negative consequences of an action into account. Consistent with a role for DA neurotransmission in processing these punishment signals, we observed, using in vivo calcium imaging, that foot shock evoked a reduction in the activity of VTA DA neurons (Fig. 6c).

To control for effects on nociception in our punishment task, we subjected the animals to a tail withdrawal test, and found this to be not affected by mesoaccumbens activation (Fig. 6d).

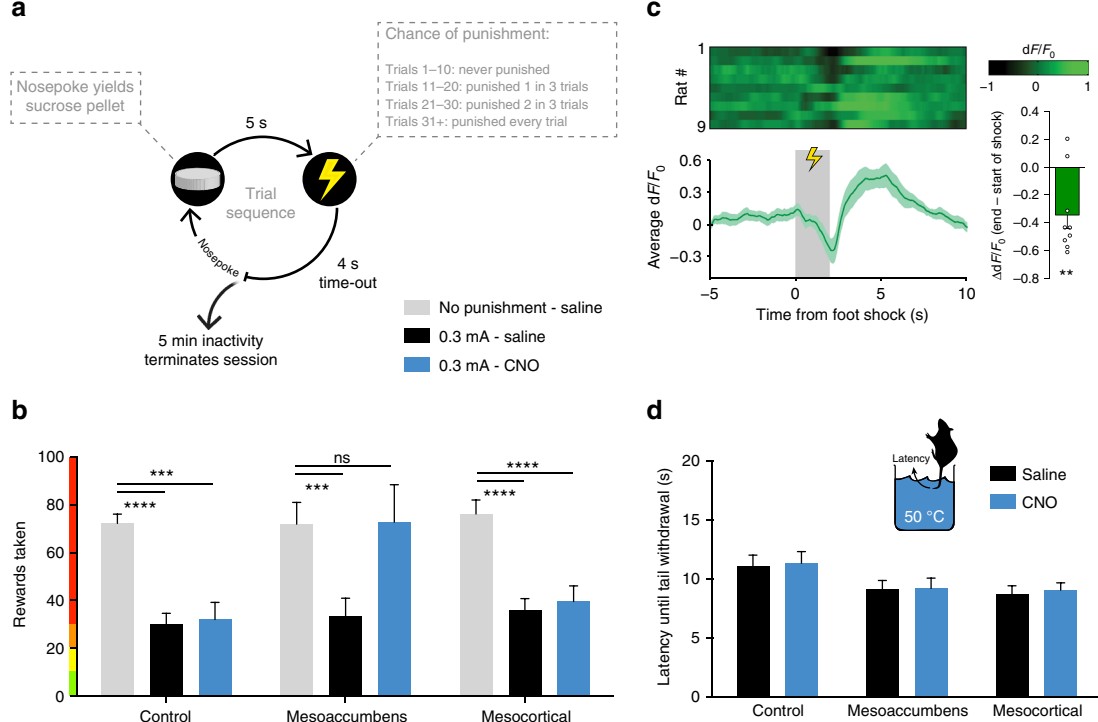

**Fig. 6** Mesoaccumbens but not mesocortical activation attenuates the effect of punishment on responding for sucrose. **a** Task design. **b** After saline treatment, foot shock punishment robustly diminished responding (Sidak's multiple comparisons test, '0.3 mA saline' versus 'no punishment saline', all $p <$ 0.001). This effect was abolished by activation of the mesoaccumbens, but not the mesocortical, pathway (Sidak's test, '0.3 mA CNO' versus 'no punishment saline' in the mesoaccumbens group, $p = 0.9995$; in mesocortical group, $p = 0.0002$; in control group, $p < 0.0001$); $n = 9$ control, $n = 9$ mesoaccumbens group, $n = 10$ mesocortical group. **c** Foot shock punishment evoked a decrease in DA neuron activity, measured using fiber photometry in TH::Cre rats (one-sample $t$-test, $p = 0.0074$, $n = 9$ rats). **d** No modulation of nociception by mesoaccumbens or mesocortical activation in the tail withdrawal test (2-way repeated measures ANOVA; main effect of CNO, $p = 0.75$; group×CNO interaction, $p = 0.99$); $n = 8$ control, $n = 9$ mesoaccumbens group, $n = 9$ mesocortical group. Data are shown as mean ± standard error of the mean; ns not significant, ****$p < 0.0001$, ***$p < 0.001$

Moreover, anxiety, as tested in the elevated plus maze (Supplementary Fig. 7a, b), was unaffected by mesoaccumbens stimulation. Consistent with the literature, we found that mesoaccumbens stimulation increased locomotion (Supplementary Fig. 8a), just like cocaine and D-amphetamine do[40,41]. We think, however, that the changes in value-based decision making observed in the punishment task, as well as in the other tasks, cannot readily be attributed to increased locomotion. First, reaction times in the punishment task were longer after mesoaccumbens activation (Supplementary Fig. 8b). Second, responding in the inactive hole in the punishment task was not changed (Supplementary Fig. 8c). Third, the effects of mesoaccumbens activation in the reversal learning task were restricted to win-stay behavior after the first reversal. Last, mesoaccumbens activation did not affect the time for the animals to complete the reversal learning session (Supplementary Fig. 8d).

**RPE processing during mesoaccumbens hyperactivity.** There are three possible explanations for the impaired negative RPE processing during mesoaccumbens hyperactivity: (1) hyperactivity of VTA DA neurons abolishes the trough in neuronal activity caused by negative reward prediction, (2) elevated DA levels lead to a baseline shift in RPE signaling, after which a decrease in DA release during negative reward prediction does not reach the lower threshold necessary to provide a learning signal in downstream regions, or (3) a combination of both.

To address the first explanation, we unilaterally injected rats with a mixture of the calcium fluorophore GCaMP6s and Gq-DREADD and tested the animals for reversal learning (Fig. 7a

and Supplementary Fig. 9). This allowed us to measure RPE signals from VTA neurons within one animal during baseline conditions and during hyperactivation of these same neurons. CNO administration did not impair the ability of VTA DA neurons to signal RPEs during reversal learning (i.e., deviations from baseline during reward prediction), inconsistent with the first possible explanation. By extension, this also excluded the third explanation. However, the second explanation is consistent with our findings that chemogenetic stimulation of the mesoaccumbens pathway increases the extracellular concentration of dopamine and its main metabolites in the NAc (Fig. 2h). Together, these data support a scenario in which the inability to adjust behavior after loss or punishment during hyperactivation of the mesoaccumbens pathway is not due to an inability of VTA neurons to decrease their firing rate during negative reward prediction, but rather by impaired processing of this learning signal within the NAc as a result of increased baseline DA levels (Fig. 7b). This observation fits well with our finding that the infusion of a DA antagonist into the NAc can prevent the effects of DREADD activation on reversal learning (Fig. 2j), a manipulation that restores the degree of NAc DA receptor activation.

## Discussion

Here, we show that hyperactivity of the mesoaccumbens pathway reduces the ability of animals to use loss and punishment signals to change behavior by interfering with negative RPE processing. Using in vivo neuronal population recordings, we show that the VTA signals reward presentation as well as reward omission

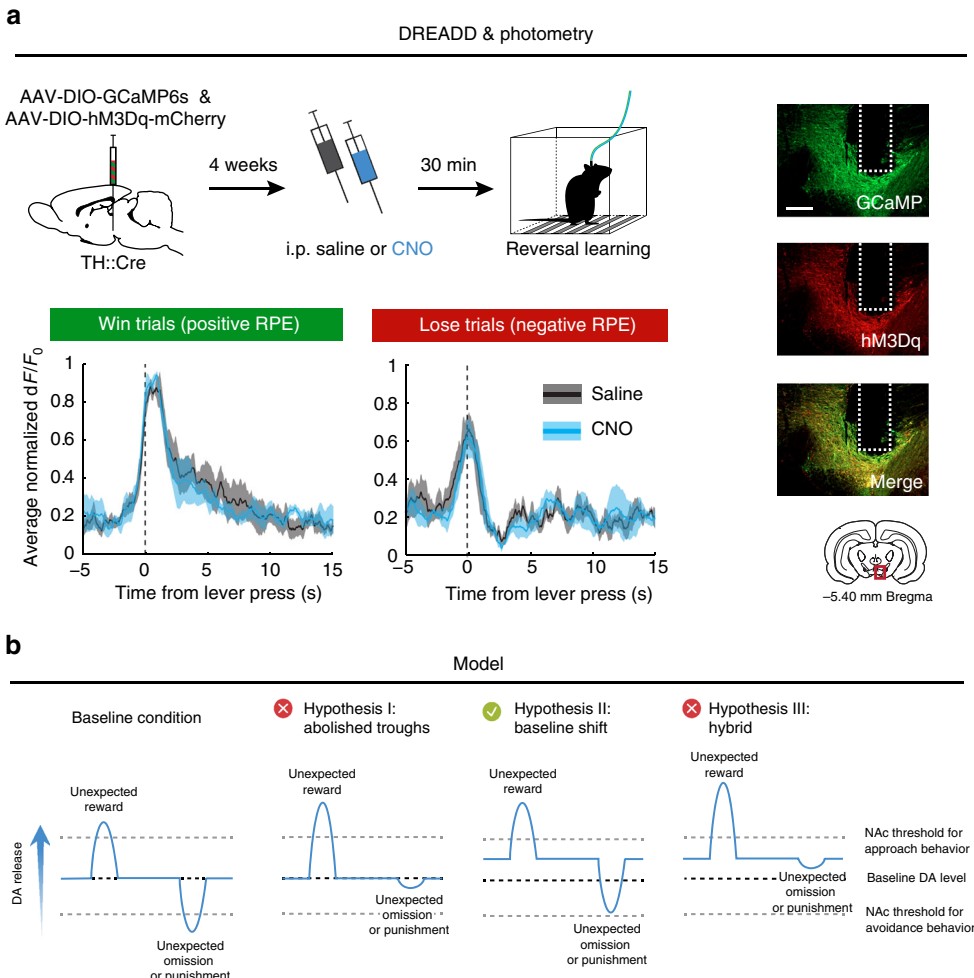

**Fig. 7** Reward prediction error processing after mesoaccumbens stimulation. **a** Animals were co-injected with GCaMP6s and Gq-DREADD and tested for reversal learning after injection of saline or CNO. VTA neurons responded in a comparable way during reversal learning after saline and CNO treatment (repeated measures in $n = 4$ animals; ANOVA, CNO×time interaction effect, win trials, $p = 0.39$; lose trials, $p = 0.38$). See Supplementary Fig. 9a for individual animals. Scale bar, 1 mm. Data are shown as mean (solid line) ± standard error of the mean (shading). **b** Proposed mechanisms: (I) hyperactivity of NAc-projecting VTA DA neurons leads to impaired coding of negative reward prediction error troughs, (II) hyperactivity shifts baseline NAc DA levels, thereby preventing the exceedance of a negative reward prediction error threshold in the NAc and impairing the ability to learn from negative feedback, or (III) a combination of I and II

during VTA neuron hyperactivity, meaning that the behavioral impairments are not caused by blunted DA neuron activity during negative reward prediction, but rather by impaired processing in the NAc as a result of elevated baseline levels of DA. Therefore, we propose a model (Fig. 7b) in which hyperactive VTA neurons signal positive and negative RPEs to the NAc, but because baseline DA tone is increased, the signaling threshold in the NAc that allows for the incorporation of negative RPEs into adaptive behavior cannot be reached during reward omission or punishment.

The majority of neurons transfected with the DREADD virus had a DAergic phenotype, chemogenetic mesoaccumbens activation replicated the effects of cocaine and D-amphetamine on reversal learning, and this effect of chemogenetic mesoaccumbens activation was prevented by intra-NAc infusion of the DA receptor antagonist α-flupenthixol. Together, this supports the notion that the behavioral changes observed in the present study are the result of chemogenetic stimulation of VTA DA cells. However, a role for non-DA VTA neurons cannot be excluded with the currently used techniques. Importantly, alongside the dense DA innervation, the VTA sends GABAergic, glutamatergic, as well as mixed DA/GABA or DA/glutamate projections to the

NAc and mPFC[16,42,43]. The role that these projections play in behavior is only beginning to be investigated, but on the basis of what is presently known, we consider it unlikely that the non-DAergic innervation of the NAc and mPFC is involved in the behavioral changes observed here. For example, optogenetic stimulation of VTA GABA neurons has been shown to suppress reward consumption, something we did not observe in our experiments[44]. In addition, by inhibiting NAc cholinergic interneurons, stimulation of VTA GABA projections to the NAc has been shown to enhance stimulus-outcome learning[45]. However, increased stimulus salience does not readily explain the deficits in reversal learning, probabilistic discounting, and punished responding for sucrose that we found in the present study. Last, stimulation of VTA–NAc glutamate neurons has been shown to produce aversive effects[46], which in our experiments most likely would have increased rather than impaired the ability to use negative feedback to alter behavior. Therefore, we think it is justified to state that the deficits in reversal learning, probabilistic discounting, and punished reward taking evoked by chemogenetic mesoaccumbens stimulation is the result of increased DA signaling in the NAc. Reversal learning impairments have previously been reported after systemic or intra-NAc treatment with

a DA $D_2$ receptor agonist in rats and humans[47–49], whereas probabilistic discounting seems to be dependent on DA $D_1$ rather than $D_2$ receptor stimulation in the NAc[50]. Together, this suggests that the behavioral effects of mesoaccumbens hyperactivity observed here rely on stimulation of both DA receptor subtypes, depending on the task structure. Interestingly, the punishment insensitivity we observed after mesoaccumbens stimulation appears inconsistent with previous studies showing that treatment with amphetamine and the DA $D_2$ receptor agonist bromocriptine make animals more sensitive to probabilistic punishment in a risky decision-making task, in which animals can choose between a small and safe reward, and a large reward with a chance of punishment[51,52]. In this latter task, however, presentation of the punishment coincides with the presentation of the large reward, and it is unknown how DA neurons respond to such an ambivalent combination of events. Importantly, risky choice behavior was found to correlate positively with DA $D_1$ receptor expression in the NAc shell[52], suggesting that the influence of NAc DA on behavior in this task may not be unidirectional.

In contrast to the mesoaccumbens projection, hyperactivity of the mesocortical pathway did not markedly affect value-based decision making. It did increase the preference for large, risky rewards over small but safe rewards in the probabilistic discounting task. However, when one of the two options yielded more sucrose reward, animals remained capable of choosing the most beneficial option, perhaps as a result of the differential roles that prefrontal $D_1$ and $D_2$ receptors play in this task[53]. That these animals maintained the capacity to make proper value-based decisions was also apparent in the reversal learning and punishment tasks. Thus, the patterns of effects of mesocortical stimulation is qualitatively different from the mesoaccumbens-activated phenotype, even though there is modest overlap, such as the increased lose-stay behavior in the probabilistic discounting task. Therefore, we do not think that the mesocortical phenotype is an attenuated version of the mesoaccumbens one, although the lower density of the mesocortical projection (Supplementary Fig. 2a) may explain the relative paucity of behavioral changes after chemogenetic mesocortical stimulation. Notably, the mesocortical pathway has been shown to be vital for certain forms of cost–benefit judgment, especially those involving uncertainty or sudden changes in task strategy[25]. As a result, manipulations of prefrontal DA affect tasks like probabilistic discounting or set shifting, but not reversal learning[25,54].

Our data emphasize the importance of balanced DA signaling in the NAc. It is reasonable to assume that brain DA concentrations are tuned to levels that are optimal to survival, and deviations from this optimum lead to the profound behavioral impairments seen in certain mental disorders. We think that our proposed model of mesoaccumbens overactivation can explain the decision-making deficits that are seen during states of increased DAergic tone, such as manic episodes, substance abuse, and DA replacement therapy in Parkinson's disease. When one cannot devalue stimuli, actions, or outcomes based on negative feedback, their value representation remains artificially elevated. Hence, outcome expectancies of choices will be unrealistically high, leading to behavior that is overconfident and overoptimistic. These inflated outcome expectancies have been demonstrated in human manic patients[2], suggesting an inability to devalue goals towards realistic levels. That this disease state is associated with abolished negative RPE signaling in the NAc is substantiated by a functional magnetic resonance imaging study in patients experiencing acute mania[55], in which activity in the NAc of manic patients remained high when monetary reward was omitted, while healthy controls showed a significant reduction in NAc activity, as expected based on RPE theory.

Most drugs of abuse enhance DA transmission in the brain, either in a direct (e.g., DA reuptake inhibition) or indirect way (e.g., disinhibition of DA neurons)[56,57]. Direct dopaminomimetics, such as cocaine and D-amphetamine, are known to mimic the symptoms of mania, such as increased arousal, euphoria, and a reduced decision-making capacity[10]. Impaired learning from negative feedback may potentially contribute to the escalation of drug use, since users may be insensitive to the thought of forthcoming negative consequences during the 'high' evoked by these drugs. Furthermore, DA replacement therapy, often prescribed to Parkinson's disease patients, has been associated with the development of problem gambling, hypersexuality, and excessive shopping behavior, a phenomenon known as the DA dysregulation syndrome[58,59]. More than a decade ago, it has already been hypothesized that these clinical features could be the result of impaired RPE learning due to 'overdosing' midbrain DA levels[30,60]. Here, we provide direct evidence to support this notion.

There is a wealth of evidence to implicate increased DA levels in harmful decision-making behavior in mental disorders[1–3]. Thus far, however, it was unknown through which pathways and by which mechanisms these effects were mediated. Here, we used behavioral tasks in rats, combined with projection-specific chemogenetics, to show that hyperactivation of the VTA leads to decision-making deficits by impairing negative feedback learning through overstimulation of NAc DA receptors. This provides a mechanistic understanding of why decision making goes awry during states of hyperdopaminergic tone, providing a possible explanation for the reckless behaviors seen during drug use, mania, and DA replacement therapy in Parkinson's disease.

## Methods

**Animals**. A total of 128 adult male Crl:WU Wistar rats (Charles River, Germany) were used for the behavioral experiments, weighing ~250 g at the start of the experiments. Rats were housed in pairs in a humidity- and temperature-controlled environment under a 12 h:12 h reversed day/night cycle (lights off at 7 am). Rats in the photometry, microdialysis, and intra-accumbens micro-infusion experiments were housed individually. Rats were food restricted (4 g of normal chow per 100 g body weight on test days, 5 g per 100 g body weight on remaining days) during the following experiments: reversal learning and probabilistic discounting. During the other behavioral tasks, animals had ad libitum access to standard chow (Special Diet Service, UK). Animals always had ad libitum access to water, except during behavioral tests. All experiments were approved by the Animal Ethics Committee of Utrecht University and conducted in agreement with Dutch laws (Wet op de Dierproeven, 1996; revised 2014) and European regulations (Guideline 86/609/ EEC; Directive 2010/63/EU).

**Surgeries**. Anesthesia was induced with an intramuscular injection of a mixture of 0.315 mg/kg fentanyl and 10 mg/kg fluanisone (Hypnorm, Janssen Pharmaceutica, Beerse, Belgium). Animals were placed in a stereotaxic apparatus (David Kopf Instruments, Tujunga, USA) and a small incision was made along the midline of the skull. Then, 1 µl of CAV2-Cre virus ($2.3 \times 10^{12}$ particles/ml) was bilaterally injected into the NAc (+1.20 mm anteroposterior (AP), ±2.80 mm mediolateral (ML) from Bregma, and −7.50 mm dorsoventral (DV) from the skull, at an angle of 10°) or the mPFC (+2.70 mm AP, ±1.40 mm ML from Bregma, and −4.90 mm DV from the skull, at an angle of 10°). The control group received a bilateral injection of 1 µl saline into the NAc. All animals received a bilateral injection of 1 µl AAV5-hSyn-DIO-hM3Gq-mCherry ($1 \times 10^{12}$ particles/ml) into the VTA (−5.40 mm AP, ±2.20 mm ML from Bregma, and −8.90 mm DV from the skull, at an angle of 10°). The viruses were infused at a rate of 0.2 µl/min. After injection, the needle was maintained at its injection position for 10 min to allow the virus to diffuse into the tissue. After surgery, the animals were given carprofen for pain relief (5 mg/kg per day for 3 days, subcutaneous (s.c.)) and saline (10 ml once, s.c.). Animals were allowed to recover for 7 days before behavioral training continued. Behavioral testing started at least 6 weeks after surgery to allow for proper viral transfection.

**Accumbens micro-infusion**. For intra-accumbens micro-infusions, 7 animals were bilaterally implanted with 26-gauge stainless steel guide cannulas (Plastics One, Raonoke, USA), 1 mm above the NAc (same coordinates as for CAV2-Cre injection, see above), after injection of the viral vectors necessary for mesoaccumbens Gq-DREADD expression. Cannulas were secured to the skull by screws and dental

cement. Injectors protruded 1 mm beyond the termination point of the guide cannulas.

Animals were habituated with saline infusions (0.5 µl/side) from 3 days before the experiment, 15 min before reversal learning training sessions. On the two experimental days, animals received infusions with saline (0.5 µl/side) or cis-(Z)-α-flupenthixol dihydrochloride (Sigma-Aldrich, Zwijndrecht, The Netherlands) dissolved in saline (10 µg dissolved in 0.5 µl/side), together with an intraperitoneal (i.p.) injection of saline or CNO 15 min prior to reversal learning. The infusion rate was set to 1 µl/min, and the injectors were left in place for an additional 30 s after the infusion was complete to allow for the diffusion of saline/flupenthixol into the brain. Between the time of infusion and testing, animals were placed back into their home cage.

**Behavioral procedures**. Animals were trained 5–7 days per week. All behavioral experiments took place between 9 am and 6 pm. The following behavioral tests were conducted in operant conditioning chambers (30.5 × 24.2 × 21.0 cm; Med Associates Inc., USA), placed within sound-attenuated cubicles: fixed ratio and progressive ratio schedule of reinforcement, reversal learning, prefeeding devaluation, probabilistic discounting and the punishment task. Testing for fixed ratio and progressive ratio of reinforcement, prefeeding devaluation and probabilistic discounting was conducted in boxes that were equipped with a sucrose receptacle flanked by two retractable levers and cue lights. The wall on the other side of the box contained a house light and tone cue generator. Testing for reversal learning and the punishment task was conducted in different boxes that contained two illuminated nose pokes, a house light, and a tone cue generator on one side of the box, and a sucrose receptacle flanked by two cue lights on the other side of the box. Sucrose pellets used were 45 mg each (SP; 5TUL, TestDiet, USA).

Chemogenetic experiments were conducted in five independent cohorts of animals:

Cohort 1: Responding for sucrose: fixed ratio 5 (FR5) schedule of reinforcement with prefeeding devaluation (with levers) (Fig. 5b), progressive ratio (PR) schedule of reinforcement (with levers) (Fig. 5c), open field (Supplementary Fig. 8a)

Cohort 2: Responding for sucrose: FR10 schedule of reinforcement (with levers) (Fig. 5a), elevated plus maze (Supplementary Fig. 7)

Cohort 3: Probabilistic discounting (with levers) (Fig. 4), reversal learning (with nose pokes) (Fig. 2), punishment task (with nose pokes) (Fig. 6)

Cohort 4: Probabilistic discounting (with levers) (Fig. 4), reversal learning (with nose pokes) (Fig. 2), elevated plus maze (Supplementary Fig. 7), tail withdrawal test (Fig. 6d)

Cohort 5: Probabilistic discounting (with levers) (Fig. 4f, g), reversal learning (with nose pokes) (Fig. 2i, j)

CNO (0.3 mg/kg dissolved in 0.3 mg/ml saline) or saline was injected i.p. 20–30 min before the start of every experiment. Unless otherwise indicated, animals were treated with CNO and saline counterbalanced between days. In between treatment days, a wash-out period of at least 48 h was used, during which behavioral training was continued.

**Fixed ratio and progressive ratio schedule of reinforcement**. Operant sessions under the FR schedule of reinforcement lasted for 1 h, during which the house light was illuminated to signal response-contingent reward availability. Animals were first trained under an FR1 schedule of reinforcement, during which pressing the active lever resulted in the delivery of one sucrose pellet, the illumination of the cue light above the active lever for 5 s and retraction of both levers. After a 10-s time-out period (during which the house light was turned off), the levers were reintroduced and the house light was turned on, signaling the start of a new trial. Pressing the inactive lever was without scheduled consequences. After acquisition of sucrose self-administration under an FR1 schedule, the response requirement was increased to FR5 (see below, prefeeding devaluation), or FR10.

Under the PR schedule of reinforcement, the response requirement on the active lever was progressively increased after each obtained reward (1, 2, 4, 6, 9, 12, 15, 20, 25, etc., see ref. [61]). A PR session ended after the animal failed to obtain a reward within 30 min. The animals were trained under FR and PR schedules before surgery. After surgery, they were retrained until we observed stable responding for at least 3 consecutive days at group level.

**Reversal learning**. Animals were trained to nose poke for sucrose under an FR1 schedule, in which responding in either of the two illuminated nose pokes resulted in the delivery of one sucrose pellet. During the reversal learning test, the nose poke holes were illuminated and responding into one of two holes (the site of the active hole was counterbalanced between animals) always resulted in reward delivery, a 0.5 s auditory tone, and switching off the nose poke lights. Responding into the inactive hole always resulted in an 8 s time-out period during which the house light and nose poke lights were turned off. A new trial began 8 s after the last response, which was signaled to the animal by illumination of the nose poke lights. When the animal made 5 correct consecutive responses in the active hole, the contingencies were reversed so that the previously inactive hole became the active one, and the previously active hole became the inactive one. The session ended when the animal completed 150 trials.

Animals had no prior experience with contingency switches before the reversal learning experiments. In between treatment days, animals were retrained on an FR1 schedule of reinforcement, in which responding of any of the two nose poke holes resulted in reward delivery. Before the intra-accumbens micro-infusions reversal learning experiments (Fig. 2i, j), animals received 8 reversal learning training sessions to gain experience with contingency changes. This was done to minimize the chance on a between-days effect on performance, i.e., a difference in performance between the first and last testing day not caused by the manipulation.

Win-stay behavior was calculated as the percentage of rewarded trials on the active nose poke hole followed by a response on that same nose poke hole in the subsequent trial. Lose-stay behavior was calculated as the percentage of non-rewarded trials on the inactive nose poke hole after which the animal responded in that same nose poke hole in the subsequent trial. Trials to criterion was defined as the total number of trials necessary to reach the first reversal (i.e., 5 consecutive responses at the active nose poke hole). Perseverative responding was defined as the total number of consecutive responses at the inactive nose poke hole directly after a reversal. For example, if after a reversal the animal chooses inactive–inactive–active, the number of perseverative responses after that reversal is 2.

**Prefeeding devaluation**. At 1 h before operant testing, animals were individually housed in standard cages where they had ad libitum access to water and standard chow (non-devalued situations) or sucrose pellets (devalued situation). The devaluation test comprised 10 min of non-reinforced lever pressing, during which pressing on either of the two levers was without scheduled consequences. This test was immediately followed by a regular session under an FR5 schedule of reinforcement. The animals were tested 4 times (devalued/non-devalued, CNO/saline), according to a within-subjects counterbalanced design. Each test day was followed by at least 2 days of regular FR5 training.

**Probabilistic discounting task**. This task was modified from refs. [33,34]. Animals were allowed to respond on a safe lever, which always yielded one sucrose pellet, and a risky lever, which yielded three sucrose pellets with a given probability. The task comprised four blocks, each consisting of 6 forced trials on the risky lever (in which only the risky lever was presented), followed by 10 free choice trials (in which both the safe and the risky lever were presented). The chance of receiving a large reward at the risky lever decreased across the four trial blocks: 100%, 33%, 16.67%, and 8.33% in blocks 1, 2, 3, and 4, respectively. Choosing the safe lever resulted in reward delivery (one pellet), a 0.5 s audio tone, and illumination of the cue light above the safe lever for 17 s. Hereafter, an intertrial interval of 3 s started, in which house- and cue lights were turned off. A rewarded response on the risky lever started the same sequence of cues, except that three sucrose pellets were delivered, with an interval of 200 ms. A non-rewarded response on the risky lever resulted in a 20 s time-out in which all lights in the operant chamber were turned off. A new trial was signaled by illumination of the house light and reintroduction of the levers. A switch of blocks was signaled to the animal by switching the house light, cue lights, and tone on and off within 2 s (1 s ON, 1 s OFF), three times in a row. This was immediately followed by the start of the forced trials sequence.

Before training on the probabilistic discounting task, animals were trained to respond on both levers, in which one lever (the future safe lever) always yielded one sucrose pellet, and the other lever (the future risky lever) always three sucrose pellets. There were 3 trial types, each with a 33.3% probability: one in which only the single-pellet lever presented, one in which only the three-pellet lever was presented, and one in which both levers were presented so the rats could choose between either lever. Hereafter, animals were trained on the probabilistic discounting task until stable task performance was observed (no significant effect of training day in a repeated-measured analysis of variance (ANOVA) over 3 days).

Win-stay behavior was calculated as the percentage of rewarded trials on the risky lever followed by a response on the risky lever in the subsequent trial. Lose-stay behavior was calculated as the percentage of non-rewarded trials on the risky lever after which the animal responded on the risky lever in the subsequent trial. Performance was calculated as the percent optimal choices in blocks 1, 3, and 4, thus percent choice for the risky lever in block 1, and percent choice for safe lever in blocks 3 and 4.

The discounting rate was calculated as follows:

$$\text{discounting rate(\% per block)} = \frac{\frac{p_{\text{block3}} + p_{\text{block4}}}{2} - p_{\text{block1}}}{3} \quad (1)$$

With $p$ being the percentage choice for the risky lever in the subscripted block; $p_{\text{block2}}$ was left out of the equation because there is no economically best choice in the second block.

**Punishment task**. Animals were placed into the operant chamber and the session started with illumination of the house light and two nose poke lights. Responding into the active nose poke hole resulted in the immediate delivery of one sucrose pellet, a 0.3 s tone cue, and illumination of the cue lights on the other side of the operant chamber, next to the sucrose receptacle. House light and nose poke lights were turned off. At 5 s after the termination of the tone cue, a second 0.3 s tone cue was played, which co-terminated with the chance of a 0.3 s, 0.3 mA foot shock. The chance of a foot shock increased across four trial blocks: trials 1–10, no

punishment; trials 11–20, 1 in 3 trials punished; trials 21–30, 2 in 3 trials punished; trials 31 and up were always punished. Cue lights were turned off after the tone–foot shock combination terminated, leaving the animals in the dark during the 5 s intertrial interval. Responding into the inactive hole was registered, but was without scheduled consequences. The session ended when no response into the active hole had been made for 5 min. Before animals were tested on the punishment task, animals were trained to nose poke for sucrose under an FR1 schedule of reinforcement (i.e., the same task, but without foot shock punishment). Between the two testing sessions, animals were retrained to respond on FR1 (without punishment) for 2 days.

**Tail withdrawal test**. This test was modified from ref. [62]. The animals were gently fixated in a towel and 3–5 cm of the tip of their tail was put in a beaker with water of $50 \pm 1$ °C. The latency until tail withdrawal was analyzed from a recorded video in a frame-by-frame manner. Animals were tested twice after CNO treatment, and twice after saline treatment (saline and CNO counterbalanced between days, with 48 h in between). The latencies of the two respective tests were averaged. When the animal did not withdraw its tail within 20 s, the animal was placed back into its home cage (this happened once in one animal).

**Elevated plus maze**. The elevated plus maze was made out of gray plexiglas, and consisted of two open arms ($50 \times 10$ cm) and two closed arms ($50 \times 10 \times 40$ cm), connected by a center platform ($10 \times 10$ cm). The maze was elevated 60 cm above the floor. Behavior was scored using Ethovision 3.0 (Noldus, Wageningen, The Netherlands). The total times spent in the closed arms, open arms, and on the central platform were analyzed. All animals received CNO and were tested once for 5 min.

**Open field test**. The open field was $100 \times 100$ cm and made out of dark plexiglas. During the 5 min test, the open field was illuminated with white light, and a white noise sound source (85 dB) was used to prevent distraction from ambient noise. Locomotor activity was measured using video tracking software (Ethovision 3.0, Noldus, Wageningen, The Netherlands). All animals received CNO and were tested once.

**Computational model**. To model the behavior of the animals in the reversal learning task, we fit the data to an extended Q-learning model. In this model, animal behavior is captured in three parameters:

$a_{win}$: learning from positive RPE (win trials)
$a_{lose}$: learning from negative RPE (lose trials)
$\beta$: the extent to which choice behavior is driven by value.

This model was chosen because it has a direct relation to midbrain dopamine by including reward prediction error factors in the equations.

On each trial, the value of left ($Q_{left}$) or right ($Q_{right}$) nose pokes was updated, depending of which of those was chosen, according to the equation:

$$Q_{s,t} = \begin{cases} Q_{s,t-1} + \alpha_{win} \cdot RPE_{t-1} & \text{for win trials} \\ Q_{s,t-1} + \alpha_{lose} \cdot RPE_{t-1} & \text{for lose trials} \end{cases} \quad (2)$$

with

$$RPE_{t-1} = \begin{cases} 1 - Q_{s,t-1} & \text{for win trials} \\ 0 - Q_{s,t-1} & \text{for lose trials} \end{cases} \quad (3)$$

in which $Q_{s,t}$ is the value of the outcome of responding into nose poke $s$ on trial $t$. Note that nose poke outcome values ranged from 0 to 1.

Nose poke outcome value at session start, $Q_{left,t=1}$ and $Q_{right,t=1}$, were set at 0.

Nose poke outcome values were converted to action probabilities using a softmax:

$$p_{s,t} = \frac{e^{\beta \cdot Q_{s,t}}}{e^{\beta \cdot Q_{left,t}} + e^{\beta \cdot Q_{right,t}}} \quad (4)$$

in which $p_{s,t}$ is the chance of choosing nose poke $s$ in trial $t$.

Best-fit model parameters were determined per animal, per session by minimizing the model's negative log likelihood $-\sum_{t=1}^{150} \log(p_{s,t})$ using MATLAB's 'fmincon' function. Each session's maximum likelihood was compared to a random choice model, in which every option had a 0.5 probability of being chosen, thus having had a log likelihood of 150 trials*log(0.5). The fit of the Rescorla–Wagner model was compared with this random choice model, both on an individual level (Supplementary Fig. 1b, Supplementary Fig. 3c), and on a group level (Supplementary Table 1), using a likelihood ratio test with the $p$ threshold set at a liberal $p = 0.1$. This type of comparison is used, since the Rescorla–Wagner model nests the chance model (chance model is a special case in the Rescorla–Wagner model in which β = 0). Although some sessions were not well explained by the Rescorla–Wagner model (i.e., animals chose randomly or used an alternative strategy; red dots in Supplementary Fig. 1b, Supplementary Fig. 3c), we decided to include all sessions in our between-treatment comparison to avoid a bias. Including only those animals in which all sessions were significantly better explained by the

Rescorla–Wagner model than by chance resulted in the same effect (i.e., a decrease in $a_{loss}$), but with higher statistical significance.

The best-fit parameters for each condition (saline, cocaine, D-amphetamine) were compared within animals using Wilcoxon matched-pairs signed rank test.

**In vivo fiber photometry**. A blue LED light (M490F2, Thorlabs, Germany) was coupled to a 400 μm core fiber optic patch cable (M76L01, Thorlabs) and connected to a fiber mount (F240FC-A, Thorlabs). It was then passed through an excitation filter (FF02-472/30-25, Semrock), reflected by a dichroic mirror (FF495/605-Di01-25 × 36, Semrock), and focused onto a 400 μm core (made from BFH48-400, Thorlabs, CF440, Thorlabs) patch cable towards the animal. For in vivo experiments, this patch cable was connected to a 400 μm implantable fiber (BFH48-400, Thorlabs) using a 2.5 mm ceramic ferrule (CF440, Thorlabs). Returning green light passed through the same patch cable onto the fiber mount. It then passed through the dichroic mirror and was deflected by a second dichroic mirror (Di02-R594-25×36, Semrock, USA) and through an emission filter (FF01-535/50-25, Semrock). The light was then focused onto a silicon-based photoreceiver (#2151 Photoreceiver, Newport Corporation, USA) using a plano-concex lens (#62-561, Edmund Optics, USA).

After photo-electron conversion, the electrical signal was pre-amplified on the photodiode ($2 \times 10^{10}$ V/A or $2 \times 10^{11}$ V/A) and then passed on to a lock-in amplifier (SR810, Stanford Research Systems). The lock-in amplifier was set to an AC grounded single input. It was then lock-in amplified in the range of 233–400 Hz, a 12 dB/oct bandwidth roll off, and a 30 or 100 ms time constant for the subsequent low-pass filtering. Sensitivity settings of the detection ranged from 1 mV to 500 mV, with normal dynamic reserve and no additional notch filters applied. The lock-in amplifier was set to the max offset (+109.21), and the phase was set to the hardware auto-adjusted value (typically in the range of 11–22°). The reference lock-in signal was translated by the hardware into TTL and coupled at 5 V to the LED controller (LEDD1B, Thorlabs) that controlled the blue LED. The lock-in amplified signal was then run onto an digitizer (Digidata 1550a Digitizer, Molecular Devices) and captured at 100 Hz − 10 kHz, typically using a 50 Hz low-pass filter. Additional TTL signals from behavioral events were simultaneously processed by the digitizer.

To correct for bleaching, raw data points $F_x$ were converted to dF/F by running-average normalization:

$$\left( \frac{dF}{F} \right)_x = \frac{F_x - F_0}{F_0} \quad (5)$$

Here, $F_0$ is the baseline, which is calculated as the average of the 50% middle values in the 30 s following every time point $F_x$.

The same surgical protocol as described above was used. Nine male TH::Cre rats (weighing 300–350 g during surgery) were used, and 1 μl of AAV5-FLEX-hSyn-GCaMP6s or AAV5-hSyn-eYFP (University of Pennsylvania Vector Core) was injected at a titer of $10^{12}$ particles/ml unilaterally into the right VTA. A 400 μm implantable fiber was lowered to 0.1 mm above the injection site and attached with dental cement. Animals were tested in the reversal learning task described above, with the difference that retractable levers were used rather than nose pokes. This was done to prevent the dopamine transients to be influenced by perseverative responses into the nose poke hole during the intertrial interval. Here, the levers remained retracted during the entire intertrial interval, so that no responses could be made until the start of the next trial. In addition, no cue lights were used and the house light was turned on continuously to prevent light contamination by the environment. Moreover, the correct responses in a row needed to obtain a reversal was set to 8 rather than 5 to increase the number of trials before the first reversal. Peri-stimulus time histograms were time-locked to the lever press (i.e., the moment of choice). In addition, 4 animals were injected with a 1 μl mixture of AAV5-FLEX-hSyn-GCaMP6s and AAV5-hSyn-DIO-hM3Gq-mCherry (both $10^{12}$ particles/ml, unilaterally in the right VTA). The AAV carrying Gq-DREADD was injected unilaterally in order to not interfere with task performance. Animals were tested in a counterbalanced fashion, so that half of the animals were first tested with saline, and the other half with CNO. In all animals expressing GCaMP6s, we tested whether a modest (0.30 mA) 2 s foot shock punishment evoked a negative RPE signal in VTA DA neurons. This was repeated 12 times in one session (with an inter-shock-interval of 40 s).

**Microdialysis**. For microdialysis experiments, 9 animals were unilaterally implanted with guide cannulas (AgnTho's, Lidingö, Sweden), 1.5 mm above the right NAc (same coordinates as for CAV2-Cre injection, see above), 4 of which also received an injection of the viral vectors necessary for unilateral mesoaccumbens Gq-DREADD expression. After 4–6 weeks, a microdialysis probe (PES membrane protruding 2 mm beyond the cannula, cut-off 15 kD; AgnTho's, Lidingö, Sweden) was placed into the guide cannula and secured. The following day, the microdialysis experiment commenced by dialysing Ringer's solution through the probe at a rate of 1 μl/min. Each sample contained 15 μl of perfusate (i.e., 15 min), which was collected in 5 μl antioxidant solution containing 0.02 N HCOOH and 0.1% cysteine HCl in milli-Q. Saline followed by CNO (1 mg/kg) was injected i.p. during dialysis.

Samples were analyzed by high-performance liquid chromatography on an Alexis 100 2D system (ANTEC Leyden, Zoeterwoude, The Netherlands), at a flow rate of 0.035 ml/min. The mobile phase consisted a solution of 2.4 mM octanesulphonic acid, 1 mM KCl, 100 mM phosphoric acid, and 15% methanol in milli-Q. Chromatograms were analyzed using Clarity software (DataApex, Prague, Czech Republic).

**Immunohistochemistry.** Animals were killed by an i.p. injection of sodium pentobarbital and perfused with phosphate-buffered saline (PBS) followed by 4% paraformaldehyde (PFA) in PBS. The brains were dissected and postfixed in 4% PFA in PBS for 24 h and then stored in a 30% sucrose in PBS solution. Brain slices (40 μm) were incubated overnight in a primary antibody solution, containing PBS with 0.3% Triton-X, 3% goat serum, and primary antibodies (1:1000) against dsRed (rabbit, Clontech 632496) and TH (mouse, Millipore MAB318). The next day, brain slices were transferred to a secondary antibody solution containing PBS with 0.1% Triton-X, 3% goat serum, and secondary goat antibodies (1:1000) against mouse (488 nm, Abcam ab150113) and rabbit (568 nm, Abcam ab175471). After an incubation period of 2 h at room temperature, slices were washed with PBS and mounted to glass slides. Histological verification was performed by a researcher unaware of the outcome of the behavioral experiment.

**Exclusion criteria.** Only animals that showed bilateral expression of hM3Gq-mCherry in the VTA were included in analyses. To exclude non-learners, animals in the probabilistic discounting task that showed a discounting rate of less than 10% per block at the end of training were excluded from the analysis.

Outlier analyses were performed on all data using the ROUT method (Q threshold set at 1.0%). Two rats were identified as outliers and removed from their respective datasets: one rat from the mesocortical group in the elevated plus maze experiment (outlier in time spent in closed arm), and one rat from the in vivo fiber photometry experiment on the basis of foot shock data (outlier in DA response to foot shock).

**Data analysis and statistics.** Data analysis and computational modeling was performed with MATLAB version R2014a (The MathWorks Inc.), statistical analyses with GraphPad Prism version 6.0 (GraphPad Software Inc.).

Statistical comparisons were made using a $t$-test for a single comparison and a (repeated measures) ANOVA was used for multiple comparisons followed by a $t$-test with Šidák's multiple comparisons correction. Paired, non-normally distributed data were compared using Wilcoxon matched-pairs signed rank test with Bonferroni correction for multiple comparisons. Welch's correction was used once, in a case where variances in the $t$-test were unequal.

Bar graphs represent the mean ± standard error of the mean, unless stated otherwise. In all figures: ns not significant, $^{\#}p < 0.1$, $^{*}p < 0.05$, $^{**}p < 0.01$, $^{***}p < 0.001$, $^{****}p < 0.0001$.

**Code availability.** Custom-written MATLAB and MedPC scripts are available upon request.

**Data availability.** The datasets generated during the current study are available from the corresponding author on reasonable request.

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

## Acknowledgements

Clozapine-N-oxide was a generous gift from the NIMH Chemical Synthesis and Drug Supply Program. We thank Roshan Cools for giving feedback on the manuscript, and the entire Adan and Vanderschuren labs for helpful discussions and feedback. This work was supported by the European Union Seventh Framework Programme under grant agreement number 607310 (Nudge-IT), The Netherlands Organisation for Health Research and Development (ZonMw) Grant 912.14.093, and The Netherlands Organisation for Scientific Research (NWO) Veni Grant 863.13.018.

## Author contributions

J.P.H.V., J.W.d.J., G.v.d.P., R.A.H.A. and L.J.M.J.V. designed the experiments. J.P.H.V., J.W.d.J., T.J.M.R., C.F.M.H., R.v.Z., M.C.M.L., G.v.d.P. and R.H. performed the experiments. J.P.H.V. analyzed the behavioral and calcium imaging data. J.P.H.V. performed and H.E.M.d.O. supervised the computational analysis. I.W. and R.H. analyzed the microdialysis experiments. J.P.H.V., H.E.M.d.O., R.A.H.A. and L.J.M.J.V. wrote the paper with input from the other authors.
