## [Peer Review File · Nature Communications]

REVIEWERS' COMMENTS:

Reviewer #1 (Remarks to the Author):

[Retracted] The authors have addressed all of my concerns, and in my estimation, also addressed the concerns of the other reviewers as well. They should be commended for adding even more confirmatory data to an already impressive data set.

I believe this set of findings provide novel insight into how different dopaminergic pathways may regulate different forms of reward-related decision making and in particular, how excessive dopaminergic activity may be an underlying cause of impairments in these processes observed in certain psychiatric disorders.

Reviewer #3 (Remarks to the Author):

The authors have thoughtfully responded to my previous concerns, and although I still have doubts about the robustness of some of their interpretations, I believe the data to be interesting aside from the interpretations they present and thus deserving of publication.

Reviewer #1:

*[Redacted] The authors have addressed all of my concerns, and in my estimation, also addressed the concerns of the other reviewers as well. They should be commended for adding even more confirmatory data to an already impressive data set.*

*I believe this set of findings provide novel insight into how different dopaminergic pathways may regulate different forms of reward-related decision making and in particular, how excessive dopaminergic activity may be an underlying cause of impairments in these processes observed in certain psychiatric disorders.*

We appreciate the constructive comments of this reviewer.

Reviewer #3:

*The authors have thoughtfully responded to my previous concerns, and although I still have doubts about the robustness of some of their interpretations, I believe the data to be interesting aside from the interpretations they present and thus deserving of publication.*

We thank the reviewer for this comment.